# The RETAIN Simulation-Based Serious Game—A Review of the Literature

**DOI:** 10.3390/healthcare8010003

**Published:** 2019-12-22

**Authors:** Simran K. Ghoman, Georg M. Schmölzer

**Affiliations:** 1Centre for the Studies of Asphyxia and Resuscitation, Neonatal Research Unit, Royal Alexandra Hospital, Edmonton, AB T5H 3V9, Canada; ghoman@ualberta.ca; 2Department of Pediatrics, Faculty of Medicine and Dentistry, University of Alberta, Edmonton, AB T6G 1C9, Canada

**Keywords:** infants, newborn, neonatal resuscitation, neonatology, serious game, simulation-based education, RETAIN, computer game, digital simulation, board game

## Abstract

*Background:* Each year, over 13 million babies worldwide need help to breathe at birth. While guidelines recommend the Neonatal Resuscitation Program course, medical errors remain common. Frequent simulation training and assessment is needed to address this competence gap; however, alternative approaches are needed to overcome barriers to access. The RETAIN (REsuscitation TrAINing) simulation-based serious game (Retain Labs Medical Inc., Edmonton, AB, Canada) may provide a solution to supplement traditional training. This paper aims to review the available evidence about RETAIN for improving neonatal resuscitation education. *Method:* Literature searches of PubMed, Google Scholar, Cochrane Central Register of Controlled Trials, CINAHL, Web of Science, and EMBASE databases were performed to identify studies examining the RETAIN serious game for neonatal resuscitation training. All of the studies describing the RETAIN board game and computer game were included. *Results:* Three papers and one conference proceeding were identified. Two studies described the RETAIN board game, and two studies described the RETAIN computer game. RETAIN was reported as usable and clinically relevant. RETAIN also improved knowledge of neonatal resuscitation by 12% and functioned as a summative assessment. Further, performance on RETAIN was moderated by players’ self-reported mindset. *Conclusion:* RETAIN can be used for the training and assessment of experienced neonatal resuscitation providers. Further studies are needed to understand the effectiveness of RETAIN to (i) improve other cognitive and non-cognitive skills, (ii) in diverse populations of neonatal resuscitation providers, (iii) in comparison to current standard training approaches, and (iv) in improving clinical outcomes in the delivery room.

## 1. Introduction

Each year, over 13 million newborns around the world require cardiorespiratory intervention at birth [1,2]. One million of these infants die, with two thirds of mortality caused by deficiencies in healthcare professionals’ (HCP) competence and communication during neonatal resuscitation [3]. To address this quality gap, the ongoing training and assessment of HCPs’ neonatal resuscitation knowledge and skills is needed to improve health outcomes [4]. 

To achieve these two goals, the guidelines recommend the simulation-based Neonatal Resuscitation Program (NRP) provider course [2]. The course requires learners to read the NRP textbook, complete four digital neonatal resuscitation simulations, and pass a multiple-choice test [5]. During the in-class portion, learners practice their skills and participate in team-based simulations, facilitated by a trained instructor [2]. However, barriers exist towards optimal efficiency and efficacy of the NRP course. First, while NRP certification requires simulation training only once every two years, HCPs experience significant decay in neonatal resuscitation competence, as soon as three months after training [6]. Frequent refresher sessions are needed to maintain HCPs’ knowledge and skills; however, traditional simulation can be prohibitively expensive, inconvenient to arrange, and potentially unstructured [7,8]. Furthermore, the robust assessment of HCPs’ preparedness to attend the NRP course, and evaluation of their neonatal resuscitation competence after attending the NRP course is currently lacking [2]. 

Simulation-based serious games may provide a solution to supplement the traditional training approach. Serious games use competition and emotional design to teach players knowledge and skills within an immersive learning environment [9]. While simulation-based serious games have been used successfully in other healthcare disciplines like surgery [10], they remain underutilized in improving access to effective training and assessment of neonatal resuscitation [11]. Evidence-based and scientifically evaluated serious games present a promising opportunity to leverage games for teaching and assessing HCPs’ neonatal resuscitation competence in complement to current approaches. We have identified the mixed-media simulation-based serious game RETAIN (REsuscitation TrAINing) to fit this description. Therefore, we aimed to review the literature about the application of the RETAIN game to improve the educational outcomes in neonatal HCPs. 

## 2. Methods

PubMed, Google Scholar, the Cochrane Central Register of Controlled Trials, CINAHL, Web of Science, and EMBASE databases were searched from database inception to 30 November 2019 to identify studies describing and/or examining the serious game RETAIN. The search terms included “RETAIN”, “neonatal resuscitation”, “resuscitation training”, “healthcare professionals”, “digital simulation”, “neonatal”, “infant”, “baby”, “serious game”, “computer game”, “board game”, “video game”, “virtual reality”, and “table-top training simulator”. No language restrictions were applied. Additionally, the reference lists of the retrieved articles were manually screened, and studies were selected based on their title, abstract, and method, if they described or used the serious game RETAIN. All relevant studies were included. 

### RETAIN (REsuscitation TrAINing)

The RETAIN game (REsuscitation TrAINing) is a learning platform for neonatal resuscitation providers to train their knowledge and skills [12]. Currently, RETAIN exists as a board game and a computer game. In RETAIN, players take on the role of a healthcare professional attending a delivery. Players use equipment pieces, monitors, and action cards to perform tasks to stabilize a newborn infant. The simulation scenarios are based on a database of real-life deliveries from a tertiary perinatal center [12]. In the board game, players train their knowledge and communication skills while working together as a collaborative team (Figure 1) [13]. In the computer game, players practice their individual knowledge and decision-making skills within the neonatal resuscitation algorithm (Figure 2) [14]. After a player performs a task, feedback is provided, including information about the infant’s visual appearance, breathing, heart rate, oxygen saturation, etc. to guide the next action. If the players perform the correct steps of the neonatal resuscitation algorithm in the correct order, the newborn infant’s health improves, and conversely deteriorates if inappropriate actions are taken. After each simulation scenario, players debrief on their performance by answering questions about the resuscitation to facilitate reflective experiential learning (e.g., Tell me in a few sentences about this case? What went well? What could have gone better? How was communication between team members? etc.) [15,16]. The intended audience for RETAIN are HCPs who may be expected to provide neonatal resuscitation care, from a variety of resource backgrounds, locations, and/or birth attendance rates [12]. 

## 3. Results

We identified three published studies and one conference proceeding describing the RETAIN game (Table 1). Two studies described the RETAIN board game, and two studies described the RETAIN computer game. All studies were of an observational design. All studies were undertaken in Edmonton, Canada. The studies investigating the educational outcomes of training with RETAIN were undertaken in a population of experienced HCPs from a tertiary perinatal center (Royal Alexandra Hospital, Edmonton, AB, Canada). None of the studies investigated any clinical outcomes as a result of training, or in comparison to another training method/control intervention. The data is presented as mean (standard deviation) or median (interquartile range). 

### 3.1. Preliminary Evaluation 

The RETAIN computer game was first presented in the literature in 2015 by Bulitko et. al, who described the process of the development of the inaugural iteration of the RETAIN computer game [18]. An early beta version of the game hosted on Aurora Toolset (BioWare, Edmonton, AB, Canada) was tested for usability by a committee of game developers (AAA and independent), academics, and computing science students from the University of Alberta (Edmonton, AB, Canada). Overall, the game was well-received, and users reported experiencing unexpected stress due to the seriousness of the game (i.e., saving the baby’s life) [18]. RETAIN was also pilot tested by two neonatologists from the Royal Alexandra Hospital (Edmonton, AB, Canada) who evaluated the game for clinical validity. The neonatologists reported the game as engaging, stressful, and interesting, while also conveying the important aspects of basic neonatal resuscitation training [18].

### 3.2. Knowledge Improvement and Knowledge Retention 

The RETAIN board game was first presented by Cutumisu et al. (2019) to evaluate knowledge improvement-related outcomes in a pre-test-post-test observational study design [13]. Researchers recruited 30 healthcare providers (e.g., neonatal registered nurses, respiratory therapists, nurse practitioners, consultants, and fellows) with NRP-provider certification to play the RETAIN board game [13]. The participants were recruited from a tertiary perinatal center and had no previous experience with RETAIN. The aim of this study was to observe the short-term knowledge retention of the neonatal resuscitation guidelines immediately after playing the RETAIN board game, measured by HCPs’ performance on a neonatal resuscitation knowledge instrument [13]. 

First, participants completed a demographic questionnaire and an open-answer neonatal resuscitation scenario to assess baseline knowledge. During the pre-test, participants explained the steps to resuscitate and stabilize the infant but received no feedback on their performance [13]. After a short tutorial, each participant led a team of two standardized players through three rounds of the RETAIN board game. All evidence-based resuscitation scenarios were moderated by a trained facilitator [13]. Lastly, participants repeated the open-answer neonatal resuscitation scenario as a post-test to measure their knowledge retention of neonatal resuscitation guidelines. Pre- and post-tests were scored using the seventh edition NRP guidelines, and were assigned one point (out of a maximum of 16) for each correct action [13]. 

The participants experienced a 12% increase in overall neonatal resuscitation performance (from 49% to 61% between the pre- and post-test, respectively) after training with the RETAIN board game [13]. Participants’ mean scores increased significantly (F (1, 29) = 21.41, *p* < 0.001, η_2_^p^ = 0.42) from the pre-test (7.87 (2.18), SE = 0.40, 95% (7.05, 8.68)) to the post-test (9.77 (2.67), SE = 0.49, 95% (8.77, 10.76)) [13]. Repeated measures ANOVA analyses reported the most knowledge gain in temperature management between the pre- and post-test (14–46%, respectively), with improved (i) placement of hat (10–43% from pre-test to post-test), (ii) plastic wrap (27–67% from pre-test to post-test), and (iii) temperature probe (7–30% from pre-test to post-test) [13]. The participants’ initiation of monitoring actions also improved (e.g., attaching pulse oximeter and electrocardiographic leads) [13].

However, while the decision to correctly provide continuous positive airway pressure (CPAP) significantly improved between the pre-test and the post-test, there was a negative effect on the task of assessing breathing from the pre-test to the post-test [13]. In addition, admission to the Neonatal Intensive Care Unit (NICU) increased between the pre-test to the post-test (0–47%, respectively) [13]. However, as none of the HCPs ended the pre-test scenario with “admission to the NICU”, these results should be interpreted with caution. Authors speculated that HCPs may have considered this action beyond the scope of the scenario and were unaware of the expectation to include this step in their answer [13]. After HCPs played RETAIN (where each scenario ended with either “admission to the NICU” or “give the baby to the mother”) they then may have realized that they were expected to end the post-test scenario with this step. Therefore, the change in performance on this step between the pre-test and post-test may not necessarily reflect improved knowledge by HCPs of NICU admission.

### 3.3. Growth Mindset-Moderated Performance

Cutumisu et al. (2018) sought to investigate whether HCPs’ performance on the RETAIN computer game was moderated by players’ self-endorsed mindset (i.e., growth mindset vs. fixed mindset) [14]. Growth mindset is the belief that effort can improve one’s abilities, in contrast to fixed mindset which endorses that one’s abilities are inherent [19,20]. Furthermore, individuals with high growth mindset perceive performance as malleable, and may be more likely to use strategies (e.g., deliberate practice) to improve their performance [19,20]. 

Cutumisu et. al (2018) recruited 50 healthcare professionals (e.g., neonatal registered nurses, nurse practitioners, respiratory therapists, consultants, and fellows) who underwent NRP provider recertification within the last 24 months to play the RETAIN computer game [14]. Participants with no previous experience with RETAIN were recruited from a tertiary perinatal center. The primary outcome of this study was to measure the number of tries participants required to complete all levels of the game, predicted by the time since participants’ last NRP course, and moderated by growth mindset [14].

Participants completed a demographic and growth-mindset questionnaire; followed by a tutorial and three rounds of the RETAIN computer game. Each round was comprised of a resuscitation scenario, which became increasingly difficult, thereby requiring a longer action sequence of wider variety to be successful [14]. Round 1 consisted of an infant whose airway needed suctioning, while Round 2 had an infant requiring mask ventilation and chest compressions; followed by Round 3 of an infant needing mask ventilation, chest compressions, and epinephrine. If a participant made four mistakes before completing each round, the simulated infant died, and the scenario ended. The measure “number of tries” describes how many attempts participants needed to successfully complete all three resuscitation scenarios [14]. 

Overall, participants required 8.47 (8.66) min to complete the game [14]. The time elapsed since NRP recertification was 8.49 (6.01) months, and the self-reported growth mindset on a pre-survey questionnaire was a combined score of 9.17 (0.93) on two 5-point Likert scale measures [14]. Cutumisu et. al reported a significant interaction between time since last NRP course and growth mindset in predicting the number of tries needed by participants (b = 0.09, SE = 0.04, beta = 0.32, t = 2.25, *p* = 0.03) [14]. Therefore, participants who completed NRP recertification more recently than their colleagues (< 8.47 months ago) completed the game in significantly more tries, but only if they endorsed lower levels of growth mindset [14]. As growth mindset increased, participants also made fewer mistakes [14]. Therefore, growth mindset moderated the relationship between participants’ task performance in RETAIN and the time since their last refresher NRP course [14]. 

### 3.4. Application as a Summative Assessment

While the prior studies reported on RETAIN as a training tool, Ghoman et al. (2019) investigated if the RETAIN board game could also be utilized as a summative assessment method of HCPs’ neonatal resuscitation competence [17]. Summative assessment evaluates learners’ competence with a final score, demonstrated by their individual performance on a particular task [21]. Researchers recruited 20 neonatal HCPs (e.g., registered nurses, nurse practitioners, respiratory therapists, and neonatal fellows) from a tertiary perinatal center, with no prior experience with the RETAIN board game [17]. The aim of this study was to examine if independent performance on the RETAIN board game could be used as an objective assessment of HCPs’ neonatal resuscitation competence [17]. 

After answering a demographic questionnaire and pre-test of an open-answer neonatal resuscitation scenario, participants underwent a tutorial and played one round of the RETAIN board game independently [17]. The game presented a scenario of a term infant with fetal bradycardia. The sessions were facilitated by a trained researcher who provided no help or feedback to participants. The sessions were video recorded, scored using the seventh edition NRP guidelines, and compared to performance in a traditional written summative assessment (pre-test) [17]. 

Overall, participants’ pre-test performance was 8.6 (2.1) out of 16 possible points (53%), and game performance was 29 (3.2) out of 40 possible points (74%). There were 10 actions shared between the pre-test and the game scenario. Out of the 10 shared actions, pre-test performance was 7.2 (1.3) (72%) and game performance was 8.8 (1.4) (88%). Authors decided on a passing score of 65%, and reported that 14/20 participants passed the pre-test assessment, while 19/20 participants passed the game-based assessment [17]. 

Participants received the game well, and reported enjoying playing the game (17/20, 85%) [17]. A total of 75% of participants reported playing between one to two hours of board games per week, while 25% of participants reported not playing board games at all during a typical week. HCPs with lower pre-test scores performed better on RETAIN if they reported more years of board game experience [17]. 

## 4. Discussion

As serious games gain traction in healthcare, it is important that evidence-based games follow a structured framework to enhance credibility, provide quality evidence of effectiveness, and overcome barriers to uptake [22]. Using a methodological approach consisting of: purposeful design, quality evaluation, and dissemination of findings, contributes to the growing body of evidence of serious games as a credible training approach for improving educational outcomes in HCPs and eventual clinical outcomes for patients [22]. While other serious games have been developed for neonatology [11], RETAIN represents a unique simulation-based serious game for neonatal resuscitation training, as is well-described in the research literature [13,14,17,18]. 

Tan and Zary (2019) developed a conceptual framework to evaluate the three components required to compose a true serious game: user experience, play, and learning [23]. User experience describes the players’ affective experiences while interacting with the game (i.e., usability, professional applicability, etc.) [23]. Play describes the game’s non-learning mechanics (i.e., within-game objectives, playable content, infrastructure, etc.) [23]. Learning describes the players’ exposure to the knowledge or skills intended to be taught by the game [23]. 

The RETAIN game encompasses all three of these elements to be considered a true serious game, as described by the outcomes of this review. First, the user experience was evaluated by game developers for usability, neonatologists for clinical relevance, and HCP participants for enjoyment measured by a post-game questionnaire [17,18]. 

Furthermore, RETAIN contains elements of play mechanics and gamification, including an objective (i.e., saving the baby’s life), playable content (i.e., action cards, equipment game-pieces, etc.), and infrastructure (i.e., emotional design to induce stress, competitive points system to rank players’ performance, etc.) [11,18]. Moreover, as players receive live feedback about the simulated infant’s health in response to the actions taken, RETAIN creates an immersive and emotional environment to generate players interest in the realistic training scenarios [11].

As RETAIN is a simulation-based serious game, learning objectives are embedded directly into the game environment. To evaluate the learning outcomes of training with RETAIN, Cutumisu et. al (2019) employed a pre-test-post-test observational study design [13]. Measuring HCPs’ incoming neonatal resuscitation knowledge allowed the authors to compare intra-participant improvement after training with RETAIN [13]. This is in contrast to another serious game, “Neonatology”, designed to teach medical students about neonatal resuscitation [24]. Swiderska et. al (2013) compared differences in learning outcomes between a control (traditional curriculum) and an intervention group (traditional curriculum plus one hour of game play) in a cohort of medical students completing their neonatal rotation [24]. While an overall improvement in neonatal resuscitation knowledge was observed, the results should be interpreted cautiously as authors did not measure participants’ baseline knowledge. 

As Cutumisu et Al. (2019) measured the HCPs’ baseline knowledge before playing the RETAIN board game, the authors observed significant improvement in knowledge retention of the neonatal resuscitation algorithm immediately after training, especially for the key steps in temperature management [13]. Therefore, training with RETAIN may improve HCPs’ preparedness to mitigate clinical outcomes such as hypothermia [25], late-onset sepsis [26], and mortality [26] in low birth-weight infants, as heat-loss prevention is currently applied ineffectively in up to 42% of clinical cases [27]. 

Furthermore, the relationship between the self-endorsed mindset and HCPs’ performance on RETAIN suggests that there may be other factors contributing to the well-characterized decline in neonatal resuscitation performance over time after training [6,14]. Understanding this relationship may highlight opportunities to build HCPs’ growth mindset as a target to improve sustained performance and efficacy of deliberate training sessions. As individuals who perceive performance as malleable rather than due to inherent ability (i.e., growth mindset) are more inclined to use strategies to improve their performance, mindset may have an important influence on educational strategies and outcomes [19]. 

Lastly, as RETAIN can be utilized for summative assessment of neonatal resuscitation performance, it presents a potential opportunity to address a variety of learning objectives including training and evaluation of learners [17]. As RETAIN incorporates debriefing at the end of each simulation scenario, using the game as an assessment method may improve HCPs’ learning experience during evaluation [28]. Therefore, RETAIN presents a promising approach to augment conventional training and improve overall access to neonatal resuscitation education. 

### Limitations

RETAIN is under rolling development, and therefore each study may have used a different iteration/prototype. As such, comparisons between the studies may be limited. In addition, all clinical studies were conducted in a similar population of experienced HCPs from a tertiary perinatal center. Therefore, the generalizability of the educational outcomes of RETAIN are currently unknown. Furthermore, while RETAIN is purported to also train non-cognitive skills such as communication, to date only knowledge-related outcomes have been studied. It is also important to note that serious games require resources (e.g., players’ access to an electronic device, time to set-up and learn to play; developers’ high cost of initial development, consistent updating, etc.) and may possibly lead to stressful experiences for players (e.g., those who do not enjoy this training method, anxiety about performance, interpersonal dynamics during team play, etc.). Finally, the effectiveness of RETAIN has not yet been evaluated in comparison to other training approaches (e.g., standard NRP training), nor demonstrated to improve HCPs’ performance in a traditional simulation, or in the delivery room. It remains inconclusive whether training with RETAIN may improve clinical outcomes for patients during neonatal resuscitation. 

## 5. Conclusions

The RETAIN board game and computer game use an evidence-based approach to develop a simulation-based serious game for the training and assessment of neonatal resuscitation competence. The board game presents a collaborative approach to improve training and assessment of neonatal resuscitation competence, while the computer game offers an opportunity for individual practice of decision-making within the neonatal resuscitation algorithm. The studies identified in this review about the computer game reported its clinical relevance and mindset-moderated performance [14,18]. Further, studies identifying the board game describe knowledge improvement after playing, and its effectiveness as a summative assessment [13,17]. Further investigation is needed to evaluate improved educational outcomes as demonstrated by improved performance on simulated and clinical neonatal resuscitation. RETAIN may provide an accessible and attractive solution to supplement conventional neonatal resuscitation education by facilitating frequent and practical training and assessment opportunities for HCPs. 

## Figures and Tables

**Figure 1 healthcare-08-00003-f001:**
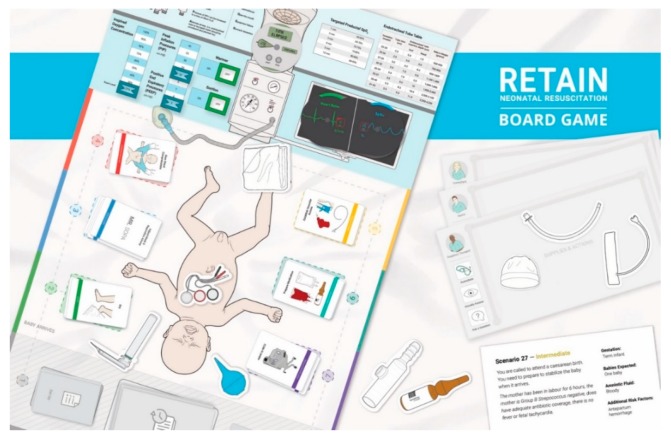
The RETAIN (REsuscitation TrAINing) board game table-top training simulator. Permission for reprint obtained from Retain Labs Medical Inc. Available online: https://playretain.com (accessed on 30 November 2019) [12].

**Figure 2 healthcare-08-00003-f002:**
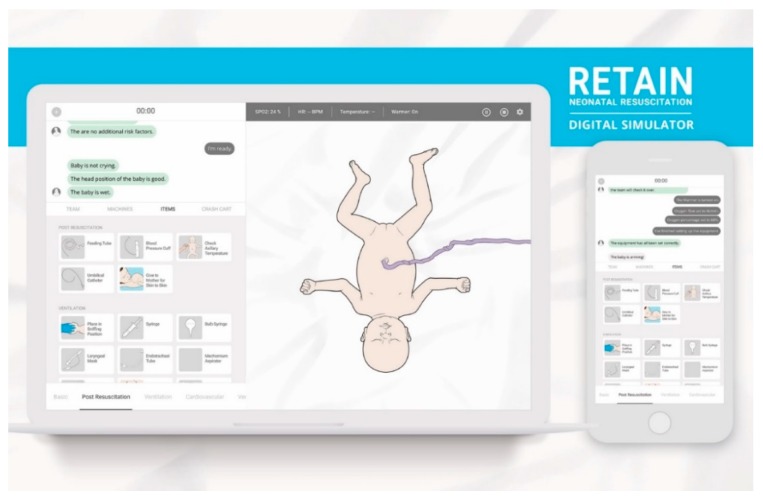
The RETAIN computer game digital training simulator. Permission for reprint obtained from Retain Labs Medical Inc. Available online: https://playretain.com (accessed on 30 November 2019) [12].

**Table 1 healthcare-08-00003-t001:** Summary of the RETAIN board game and computer game. Healthcare professional is abbreviated as HCP. Neonatal resuscitation program is abbreviated as NRP.

Reference	Participants	Objectives	Study Design	Outcome
**Board Game**				
Cutumisu et. al 2019 [13]	30 neonatal HCPs	Improves short-term knowledge retention	Compared HCPs’ performance on a written simulation scenario before and after playing the game	HCPs knowledge of the correct steps of neonatal resuscitation improved by 12%; biggest improvement in steps of temperature management
Ghoman et. al 2019 [17]	20 neonatal HCPs	Functional as a summative assessment tool	Compared HCPs’ performance on a written simulation scenario to performance on the game	HCPs performed better on the game compared to a traditional assessment, and enjoyed
**Computer Game**				
Bulitko et. al 2015 [18]	Game developers, computer science students, and two neonatologists	Development and Pilot testing	Preliminary observational data collection and pilot play testing sessions of the game	The game was reported as well-received, eliciting stress, clinically valid, engaging, and useful for basic neonatal resuscitation training
Cutumisu et. al 2018 [14]	50 neonatal HCPs	Effect of mindset in moderating performance	Analyzed HCPs’ time elapsed since NRP-training, self-reported growth mindset, and performance on the game	Time since last NRP course predicted number of tries needed to successfully complete the game when moderated by mindset; HCPs with higher growth mindset made fewer mistakes

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
