# Peer review of "The RETAIN Simulation-Based Serious Game—A Review of the Literature"

_healthcare, 2019, doi:10.3390/healthcare8010003_

Round 1
Reviewer 1 Report
Thank you for giving me an opportunity to review this manuscript, which reports a systematic review on the RETAIN simulation-based serious game for neonatal resuscitation. The authors selected a total of 4 studies published from data inception to November 30, 2019. They addressed the importance of RETAIN for training and assessment of experienced neonatal resuscitation providers. As this manuscript was well written, I have no suggestions to be revised. Thank you again for their nice manuscript.
Author Response
We thank the reviewer for their recommendation for publication and thoughtful comments.
Reviewer 2 Report
The RETAIN game is a learning platform dedicated to training healthcare professionals in neonatal resuscitation. This manuscript paper aims to review the literature on the application of this game in improving the educational outcomes of neonatal health professionals. The methodology employed in RETAIN places the participant at the center of their training, developing a critical sense of what is learned and acquiring skills to relate this knowledge to their daily lives. This approach containing purposeful design, quality evaluation, and dissemination of findings contributes to consolidate the evidence that serious games are a valid training approach to improve the educational outcomes of health professionals.
Broad comments
The manuscript has a good quality presentation, and highlight in exposes and interpret works on validation of the game RETAIN. Arouses interest in readers in the area of neonatal emergencies and may extend to pediatrics in general. It also catches the eye of gamification-related professionals, and alternative educational strategies, areas that are on the rise.
The approach synthesizes several works by highlighting positives and negatives points in an organized way. At the same time, it analyzes and discusses its results compared with each other. This smooth the work of researchers interested, as well as inserting a point of view on the topics.
The introduction contextualizes the current scenario and clarifies the purpose of the work. But it must be said that technologies like RETAIN are a complement to conventional training.
The methodology presented is technically correct and well done. Few papers found, yet they are robust enough to have a thorough discussion and draw conclusions.
The results are clearly described and discussed. The conclusion should be improved as it needs to better relate to the topics displayed in the results, and contain more detail. This way, the work will acquire more importance.
The advantages of the board game and the PC game presented in the description of figures 1 and 2 should be added to the conclusion as it clarifies when to use one or the other and reinforces the results.
The manuscript reports that the player receives feedback from his tasks. Perform the debriefing using the feedback in cyclical education simulations is interesting. Debriefing makes summative assessments more didactic. The references should be revised. Report and discuss the debriefings of RETAIN Simulation if the review contains subject information.
Real-time feedback on the newborn's situation while performing tasks helps in the realism of the simulation and generate interest in the game. This information should also be included considering the references.
Specific comments
Line 43 says: “However, this approach is inefficient and ineffective.”. The affirmation is false. The NRP course methodology is efficient and appropriate for its purpose. RETAIN is an alternative that can improve and innovate the way traditional courses (which do not use technology in their simulations) are taught. Even though RETAIN has shown better results, the statement cannot be said. The justifications given can only use as weaknesses of the NRP course.
Line 47 cites a nonstandard simulation. Scenarios are had standard according to current guidelines and their objectives. Standardizing simulations makes no sense, as it mischaracterizes the spontaneous character of the participants.
In line 266 should be cited only in the studies reported in the review, so as not to generate doubts.
Author Response
The RETAIN game is a learning platform dedicated to training healthcare professionals in neonatal resuscitation. This manuscript paper aims to review the literature on the application of this game in improving the educational outcomes of neonatal health professionals. The methodology employed in RETAIN places the participant at the center of their training, developing a critical sense of what is learned and acquiring skills to relate this knowledge to their daily lives. This approach containing purposeful design, quality evaluation, and dissemination of findings contributes to consolidate the evidence that serious games are a valid training approach to improve the educational outcomes of health professionals.
Broad comments
The manuscript has a good quality presentation, and highlight in exposes and interpret works on validation of the game RETAIN. Arouses interest in readers in the area of neonatal emergencies and may extend to pediatrics in general. It also catches the eye of gamification-related professionals, and alternative educational strategies, areas that are on the rise.
The approach synthesizes several works by highlighting positives and negatives points in an organized way. At the same time, it analyzes and discusses its results compared with each other. This smooth the work of researchers interested, as well as inserting a point of view on the topics.
Thank you sincerely for your comments.
The introduction contextualizes the current scenario and clarifies the purpose of the work. But it must be said that technologies like RETAIN are a complement to conventional training.
Thank you for this excellent comment. This has been added to the manuscript in the abstract, introduction, discussion, and conclusion.
The methodology presented is technically correct and well done. Few papers found, yet they are robust enough to have a thorough discussion and draw conclusions.
The results are clearly described and discussed. The conclusion should be improved as it needs to better relate to the topics displayed in the results, and contain more detail. This way, the work will acquire more importance.
The advantages of the board game and the PC game presented in the description of figures 1 and 2 should be added to the conclusion as it clarifies when to use one or the other and reinforces the results.
Thank you for these comments to strengthen the conclusion. This has been added to the manuscript in the conclusion.
The manuscript reports that the player receives feedback from his tasks. Perform the debriefing using the feedback in cyclical education simulations is interesting. Debriefing makes summative assessments more didactic. The references should be revised. Report and discuss the debriefings of RETAIN Simulation if the review contains subject information.
Thank you for this comment. This has been added to the methods, and to the discussion. The references have been revised to cite experiential learning cycles and debriefing in serious games. However, the papers included in this review did not present results from debriefing within the RETAIN simulation. Therefore, we are unable to report on the debriefings of the RETAIN simulation in a meaningful, evidence-based way.
Real-time feedback on the newborn's situation while performing tasks helps in the realism of the simulation and generate interest in the game. This information should also be included considering the references.
Thank you for this helpful comment. This has been added to improve the discussion.
Specific comments
Line 43 says: “However, this approach is inefficient and ineffective.”. The affirmation is false. The NRP course methodology is efficient and appropriate for its purpose. RETAIN is an alternative that can improve and innovate the way traditional courses (which do not use technology in their simulations) are taught. Even though RETAIN has shown better results, the statement cannot be said. The justifications given can only use as weaknesses of the NRP course.
Absolutely correct, thank you for this comment. The sentence from Line 43 has been modified, and clarification of RETAIN as an alternative to improve rather than replace traditional courses has been added to the abstract, introduction, and discussion.
Line 47 cites a nonstandard simulation. Scenarios are had standard according to current guidelines and their objectives. Standardizing simulations makes no sense, as it mischaracterizes the spontaneous character of the participants.
We thank the reviewer for this comment, and agree that the phrase "standardized simulations" is imprecise to convey our intent behind this sentence. We have replaced "unstandardized" with "unstructured" in Line 47, and hope this modification is acceptable.
Our intent with the phrase "unstructured" is to refer to traditional simulation being directed at the discretion of the instructor (relies on a trained and experienced instructor who can successfully facilitate and adapt a realistic scenario). This is contrast to the RETAIN simulation scenarios which are evidence-based (clinical history, heart rate, oxygen saturation, etc.,... are based on a database of real-life deliveries from a tertiary perinatal centre) and are structured (the game contains a decision-tree with feedback provided for any potential action taken by players during the simulation). We wish to convey that the RETAIN simulation is still functional even when facilitated by an instructor with less expertise than would be required by traditional "unstructured" simulation.
If this modification remains unacceptable to the reviewers, the phrase "potentially unstructured" can be deleted from the manuscript in the introduction.
In line 266 should be cited only in the studies reported in the review, so as not to generate doubts.
Thank you for this comment. We have adapted the conclusion and have added the citations for the studies reported in the review as suggested.
Reviewer 3 Report
In this manuscript, Simran K. Ghoman and Georg M. Schmölzer aimed to review the literature about the application of the RETAIN game to improve educational outcomes in neonatal HCPs. They concluded that RETAIN can be used for training and assessment of experienced neonatal resuscitation providers. Overall, the elements of the study are well-organized and scientifically coherent, but need a minor revision.
Details follow.
Line 142: information about the “Admission to NICU” are needed
Line 175: rephrased as “growth mindset moderated the relationship between participants’ task performance in RETAIN and the time since their last refresher NRP course”
Line 256: RETAIN games can also lead to stressful experiences for players due to the serious content of scenarios, anxieties about performance or interpersonal dynamics between players. They have limitations such as accessibility (eg, access to a computer, smartphone or tablet), high cost of initial development and consistent updating.
Author Response
In this manuscript, Simran K. Ghoman and Georg M. Schmölzer aimed to review the literature about the application of the RETAIN game to improve educational outcomes in neonatal HCPs. They concluded that RETAIN can be used for training and assessment of experienced neonatal resuscitation providers. Overall, the elements of the study are well-organized and scientifically coherent, but need a minor revision.
Thank you for your recommendation for publication, and thoughtful feedback.
Details follow.
Line 142: information about the “Admission to NICU” are needed
Thank you for this important comment. Information from the Cutumisu et al 2019 paper about “admission to the NICU” has been added to the manuscript to the results section.
Line 175: rephrased as “growth mindset moderated the relationship between participants’ task performance in RETAIN and the time since their last refresher NRP course”
Excellent comment, the sentence has been rephrased as suggested.
Line 256: RETAIN games can also lead to stressful experiences for players due to the serious content of scenarios, anxieties about performance or interpersonal dynamics between players. They have limitations such as accessibility (eg, access to a computer, smartphone or tablet), high cost of initial development and consistent updating.
Thank you for this thoughtful comment. This information strengthens the manuscript, and has as such been added to the Limitations section as suggested.